# Incidence and Characteristics of Endophthalmitis after Cataract Surgery in Poland, during 2010–2015

**DOI:** 10.3390/ijerph16122188

**Published:** 2019-06-20

**Authors:** Michał S. Nowak, Andrzej Grzybowski, Katarzyna Michalska-Małecka, Jacek P. Szaflik, Milena Kozioł, Wojciech Niemczyk, Iwona Grabska-Liberek

**Affiliations:** 1Provisus Eye Clinic, Czestochowa 42-209, Poland; 2Saint Family Hospital Medical Center, Lodz 90-302, Poland; 3Department of Ophthalmology, University of Warmia and Mazury, Olsztyn 10-082, Poland; ae.grzybowski@gmail.com; 4Institute for Research in Ophthalmology, Foundation for Ophthalmology Development, Poznan 60-554, Poland; 5Department of Ophthalmology, School of Medicine in Katowice, Medical University of Silesia, Katowice 40-952, Poland; k.michalska.malecka@gmail.com; 6Department of Ophthalmology, SPKSO (Samodzielny Publiczny Kliniczny Szpital Okulistyczny) Ophthalmic Hospital, Medical University of Warsaw, Warsaw 03-709, Poland; jacek@szaflik.pl; 7Department of Analyses and Strategies, Polish Ministry of Health, Warsaw 00-952, Poland; m.koziol@mz.gov.pl (M.K.); w.niemczyk@mz.gov.pl (W.N.); 8Department of Ophthalmology, Centre of Postgraduate Medical Education, 231 Czerniakowska str., Warsaw 01-416, Poland; iliberek@gmail.com

**Keywords:** cataract surgery, acute endophthalmitis, chronic endophthalmitis

## Abstract

*Background*: The assessment of the incidence and characteristic of acute and chronic postoperative endophthalmitis (POE) after cataract surgery in Poland during 2010–2015. *Patients and methods:* All hospitalizations of patients, in the National Database of Hospitalizations, who underwent cataract surgery alone or in combined procedures in Poland between January 2010 and December 2015, with a billing code of endophthalmitis, were selected. Acute endophthalmitis was identified if symptoms occurred within 1–42 days from the cataract surgery and chronic endophthalmitis if symptoms occurred ≥43 days after cataract surgery, respectively. *Results:* In total, 1331 cases of POE after 1,218,777 cataract extractions were identified. The overall incidence of POE decreased from 0.125% in 2010 to 0.066% in 2015. In multiple logistic regression analyses, increasing age was significantly associated with acute POE, while type II diabetes mellitus, extracapsular cataract extraction, and one-day surgery were significantly associated with chronic POE. The combined cataract surgery and male sex were significant risk factors for both acute and chronic POE. A total of 62.51% of all eyes affected by POE received antibiotic treatment and 37.49% had vitrectomy treatment. *Conclusions:* During the study period, the total incidence of postoperative endophthalmitis after cataract surgery decreased significantly.

## 1. Introduction

Postoperative endophthalmitis is a serious complication of cataract surgery, with the incidence varying from 0.02% to 0.71% according to recent studies. These studies also revealed a significant decline in the incidence of postoperative endophthalmitis in the twenty-first century [1,2]. The results of the European Association of Cataract and Refractive Surgeons (ESCRS) report showed that the use of intracameral cefuroxime at the end of the cataract surgery significantly reduced the occurrence of postoperative endophthalmitis as well as improved the surgery techniques with the use of injectable lenses, topical anesthesia, and microincisions [3]. According to this report, the existing risk factors for postoperative endophthalmitis included clear corneal incisions, the use of silicone intraocular lenses, and the presence of surgical complications [3]. Although the ESCRS endophthalmitis report was based on a multicenter, international study and comprised 16,603 participants, its results were published over a decade ago [3].

Globally, cataract surgery is the most common ocular surgery; however, reports on the incidence of endophthalmitis after cataract on a national level are limited [4,5]. In 2013, Friling et al. [6] reported the incidence of endophthalmitis in the Swedish National Cataract Register from 2005 to 2010. In 2016, Creuzot-Garcher et al. [7] reported the incidence of acute postoperative endophthalmitis in France from 2005 to 2014. Two studies from Canada [8,9] published data from the State Control for Health Insurance Plan regarding all cataract surgeries performed during 1996–2005 in Quebec province and during 2002–2006 in Ontario province. The aim of the present study was to assess the incidence and characteristic of endophthalmitis after cataract surgery in the overall population of Poland from 2010 to 2015. Our study was part of a project titled “Maps of Healthcare Needs—Database of Systemic and Implementation Analyses,” which was co-financed by the European Union funds through the European Social Fund under the Operational Program Knowledge, Education, and Development [10].

## 2. Materials and Methods

In Poland, the National Health Fund (Narodowy Fundusz Zdrowia—NFZ) maintains the national database of hospitalizations, which records all medical procedures in public and private hospitals financed from public sources. The national database of hospitalizations provides accurate population-based medical data, which include the diagnoses coded according to the International Classification of Diseases, 10th Revision (ICD-10), and all procedures performed, coded using the International Classification of Diseases, 9th Revision (ICD-9), procedure codes and unique NFZ codes corresponding to certain hospital procedures. It also compiles the socio-demographic data of all patients including personal identification number (PESEL), age, sex, and place of residence. Our study design was a population-based retrospective epidemiological survey. The subject sampling method was published in our previous studies. In brief: “The data from the national database of hospitalizations from all patients who underwent cataract surgery alone or in combined procedures in Poland between January 2010 and December 2015 were assessed [11]. For each individual patient, cataract surgery alone or as a combined procedure with corneal transplantation, glaucoma filtration surgery, or vitrectomy was retrospectively identified. The ICD-9 code 13.4 was used to identify cataract extraction performed by phacoemulsification, with 13.2, 13.3, and 13.5 codes used to identify other types of cataract extractions. The following NFZ codes were used: B12–B15, B18, and B19 with regard to cataract surgery alone; B04–B06 with regard to cataract surgery combined with corneal transplantations; B11 with regard to cataract surgery followed by glaucoma filtration surgery; B16 and B17 with regard to cataract surgery combined with vitrectomy.” The ICD-10 codes H44.0 and H44.1 were used to identify endophthalmitis. All hospitalizations of patients who underwent cataract surgery alone or in combined procedures in Poland during the researched period, with a billing code of endophthalmitis, were selected. Acute endophthalmitis was diagnosed if the symptoms occurred within 1–42 days after cataract surgery and chronic endophthalmitis was diagnosed if the symptoms occurred ≥43 days after cataract surgery. Patients with diabetes mellitus (DM) were identified with ICD-10, E10, and E11 codes and received DM medication before cataract surgery. Population data were obtained from the Central Statistical Office of Poland (Głowny Urzad Statystyczny—GUS) [12].

The statistical analysis included the annual volume of cataract surgery, calculations of incidence of both acute and chronic endophthalmitis, and the demographic and surgical characteristics of patients with endophthalmitis (the socio-demographic data including age, sex, and place of residence were anonymously recorded). Independent Wald tests were used for risk factor analysis. Multiple logistic regression statistics were used to investigate the association of endophthalmitis with several risk factors, including age, gender, rural residence, one-day procedure, combined surgery, extracapsular cataract extraction surgical technique, surgery in a non-multidisciplinary hospital and the presence of diabetes mellitus. Odds ratios (ORs) were computed. *p*-values less than 0.05 were considered statistically significant. The study protocol adhered to the tenets of the Declaration of Helsinki for research involving human subjects and was approved by the Polish Ministry of Health. The Polish Ministry of Health is entitled to process the National Health Fund’s data by the law of Republic of Poland, so we do not have any ethical approval number.

## 3. Results

Cataract surgery and postoperative endophthalmitis: Table 1 shows the total number of postoperative endophthalmitis (POE) cases and the total number of all cataract surgeries performed in Poland alone or as combined procedures with vitrectomy, glaucoma filtration surgery, and corneal transplantation, between January 2010 and December 2015, matched with population data by age group. In the researched period, the number of cataract surgeries in Poland increased by 17.9% from 201,083 cases in 2010 to 237,098 cases in 2015, with a significant decrease in the years 2011–2013. In total, 1,218,777 cataract extractions (alone or combined with other procedures) were performed during 2010–2015.

Among these cataract surgeries, 1331 were associated with postoperative endophthalmitis (POE), including 584 cases of acute POE (within 42 days from cataract surgery) and 747 chronic POE (≥42 days after cataract surgery) (Table 2). The overall incidence of POE decreased from 0.125% in 2010 to 0.066% in 2015 (with the mean incidence of 0.109%). The incidences of acute and chronic POE decreased from 0.047% and 0.078% in 2010 to 0.035% and 0.031% in 2015, respectively (Table 2). The differences between the incidences of acute, chronic, and the total number of POE during 2010–2015 were statistically significant (*p* = 0.0018, *p* = 0.0147, and *p* = 0.0000, respectively) with a temporal increase in the incidence of POE during 2011–2013.

The demographic and surgical characteristics of all cases of POE are presented in Table 3 and Table 4. The mean age of subjects with acute POE was 71.2 ± 11.9 years and for those with chronic POE was 73.7 ± 12.1 years. In Poland, 28.85% of all cases of POE were identified in rural residents and 42.30% in men. Diabetes mellitus type I and type II were diagnosed in 4.43% and 18.78% of subjects with postoperative endophthalmitis, respectively. The extracapsular cataract extraction surgical technique was used in 5.94% of cases and 32.61% were identified after one-day cataract surgery. Postoperative endophthalmitis occurred in 137 eyes (10.30%) after cataract surgery combined with pars plana vitrectomy, in 20 eyes (1.50%) after cataract surgery combined with glaucoma filtration surgery, and in 16 eyes (1.20%) after cataract surgery combined with corneal transplantation. Young males had relatively more POE probably due to traumatic cataract. A total of 62.51% of all eyes affected by POE in Poland during 2010–2015 received antibiotic treatment and 37.49% had vitrectomy treatment (with a rate of 44.86% among the cases of acute POE) (Figure 1).

### Multiple Logistic Regression Modeling

Multivariate logistic regression models were constructed to analyze the risk factors for acute and chronic postoperative endophthalmitis (POE) after cataract surgery in Poland during 2010–2015 (Table 5). The sensitivity/specificity evaluation of multiple logistic regression models for acute and chronic postoperative endophthalmitis with receiver operating characteristic (ROC) curves are presented in Figure 2 and Figure 3. Our analysis showed that acute POE was significantly associated with age (OR 0.99, 95% confidence interval (CI) 0.98–1.00) and male sex (OR 1.35, 95% CI 1.11–1.64). Chronic POE was also significantly associated with male sex (OR 1.28, 95% CI 1.08–1.53) as well as with type II diabetes mellitus (OR 1.42, 95% CI 1.18–1.75), extracapsular cataract extraction (OR 3.09, 95% CI 2.20–4.23), and one-day surgery (OR 0.75, 95% CI 0.61–0.91). The combined cataract surgery was a significant risk factor for both acute and chronic POE (OR 4.25, 95% CI 3.09–5.74 and OR 4.41, 95% CI 3.38–5.67, respectively). No association was found between acute and chronic POE with surgery in a non-multidisciplinary hospital or with rural residence of our study subjects.

## 4. Discussion

Our study provides, for the first time, data concerning the incidence and characteristics of postoperative endophthalmitis (POE) after cataract surgery in Eastern Europe. The study evaluates the trends of acute and chronic POE after cataract surgery in the overall population of Poland in the years 2010–2015. During the study period, the total number of all POE cases decreased from 252 in the 2010 to 157 in the 2015, while the total number of cataract surgeries increased from 201,083 cases in the 2010 to 237,098 cases in 2015. The overall incidence of POE decreased from 0.125% in 2010 to 0.066% in 2015, with a significant decrease in the incidences of both acute and chronic POE.

Our findings are in agreement with the results of recent studies from other countries, which showed a significant decrease in the incidence of POE after cataract surgery over time [6,7,13,14,15]. The results of the recent studies of POE which comprised over 100,000 participants are presented in Table 6. Our total incidence rate of POE (0.109%) was similar to the rate found in a nationwide study in France [7]. It was lower than the rates found in Canada and the Medicare Database in USA [8,9,13] and was higher than the rates found in Malaysia, Sweden, and in members of Kaiser Permanente, California, USA [6,14,16]. But these studies recorded only the incidence of acute postoperative endophthalmitis after cataract surgery. Our total incidence rate of acute POE was rather low (0.048%) and was the second lowest among nationwide studies after the incidence rate in Sweden. In Poland, the recorded incidence rate of chronic POE in the years 2010–2015 (0.061%) was higher than that of acute POE. However, during the study period it was reversed and, in the year 2015, the incidence of acute POE was higher than the incidence of chronic POE. Direct comparison of our results to the results obtained in studies from Iran and India [2,17] is limited due to the difference in study design. Although they comprised over 100,000 participants, those two studies were single-hospital studies. Between the years 2010 and 2015, no simultaneous bilateral endophthalmitis was officially reported in Poland, since the legal regulations for immediate sequential bilateral cataract surgery (ISBCS) were introduced in January 2017. However, the results of recently published studies on ISBCS revealed that when the guidelines for strict separation of the two surgical procedures were followed, as well as after separate bilateral cataract surgeries less than five days apart, no simultaneous bilateral endophthalmitis was detected [18,19]. We analyzed the surgical characteristic of cataract surgery in Poland to find the possible explanation for the acute and chronic POE decrease over the years. We found that during the study period the total number of one-day cataract surgeries significantly increased, while the use of extracapsular lens extraction significantly decreased [10].

In our study, increasing age was significantly associated with acute POE, while type II diabetes mellitus, extracapsular cataract extraction, and one-day surgery were significantly associated with chronic POE. The combined cataract surgery and male sex were significant risk factors for both acute and chronic POE. Our findings were in agreement with the results of some previous studies, which revealed that acute POE was significantly associated with older age, male gender, black race, diabetes mellitus, presence of renal disease as well as extracapsular cataract extraction, cataract surgery combined with other procedures, intraoperative posterior capsule rupture, and non-use of intracameral antibiotic [2,6,7,13,14,16,20,21].

Although intracameral cefuroxime (Aprokam, Laboratoires Thea, Clermont-Ferrand, France) was commercially available in Poland since 2012, we did not include the intracameral antibiotic injection into the multiple regression analysis of endophthalmitis risk factors. This procedure was not widely used during the study period due to the significant reduction in the reimbursement cost of cataract surgery by the National Health Fund. This cost reduction was also a possible factor of the temporary increase in the incidence of POE in the years 2011–2013.

There are some limitations related to the present study. The major limitation is the possible presence of misclassification. The diagnosis of endophthalmitis was based on clinical presentation rather than a stricter bacteriological definition. Other limitations include errors in using specific ICD-10, ICD-9, and NFZ codes. However, such mistakes likely had only a minor impact on the study findings, because the present study was country based and covered the overall population of Poland. During the study period, the use of an intracameral antibiotic injection at the end of the cataract surgery was not officially reported in the NFZ national database of hospitalizations as well as other possible risk factors such as surgical complications and the type of intraocular lens (IOL) used. However, the cost-effectiveness of the use of intracameral antibiotics in reduction of the substantial costs associated with the treatment of POE after cataract surgery is well known [22].

## 5. Conclusions

In summary, our study showed, for the first time, the incidence of acute and chronic POE after cataract surgery in the overall population of Poland in the years 2010–2015 as well as the existing risk factors. During the study period, the total incidence of postoperative endophthalmitis after cataract surgery decreased significantly, while the total the number of cataract surgeries significantly increased. Globally, large population-based data regarding the incidence rate of chronic POE are still lacking. The present study is the first nationwide study which reports the prevalence of chronic POE after cataract surgery.

## Figures and Tables

**Figure 1 ijerph-16-02188-f001:**
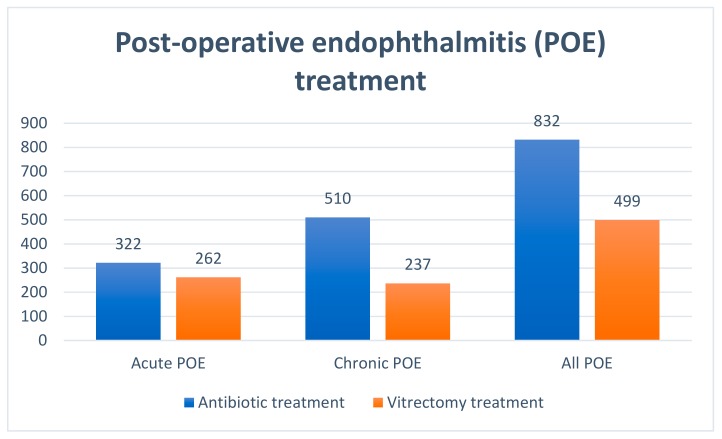
Treatment of postoperative endophthalmitis (POE) after cataract surgery in Poland during 2010–2015.

**Figure 2 ijerph-16-02188-f002:**
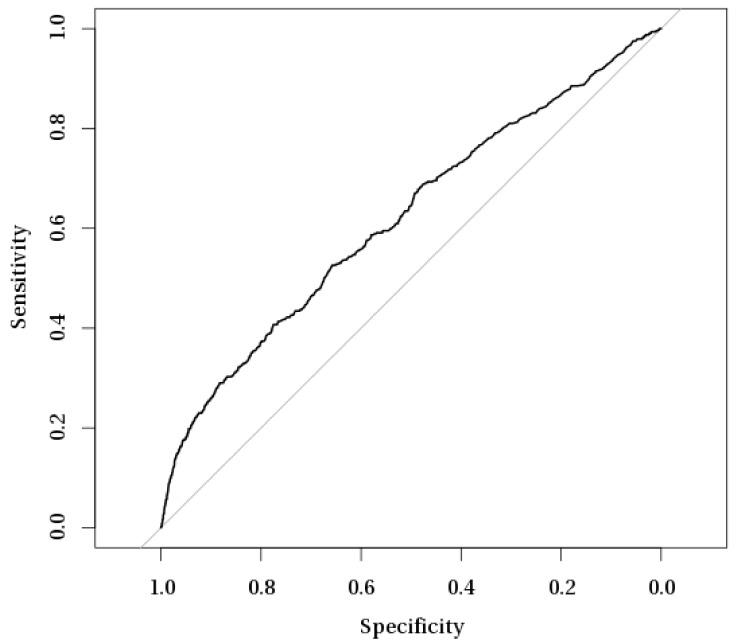
Receiver operating characteristic (ROC) curve analysis of sensitivity/specificity evaluation of multiple logistic regression model for acute postoperative endophthalmitis.

**Figure 3 ijerph-16-02188-f003:**
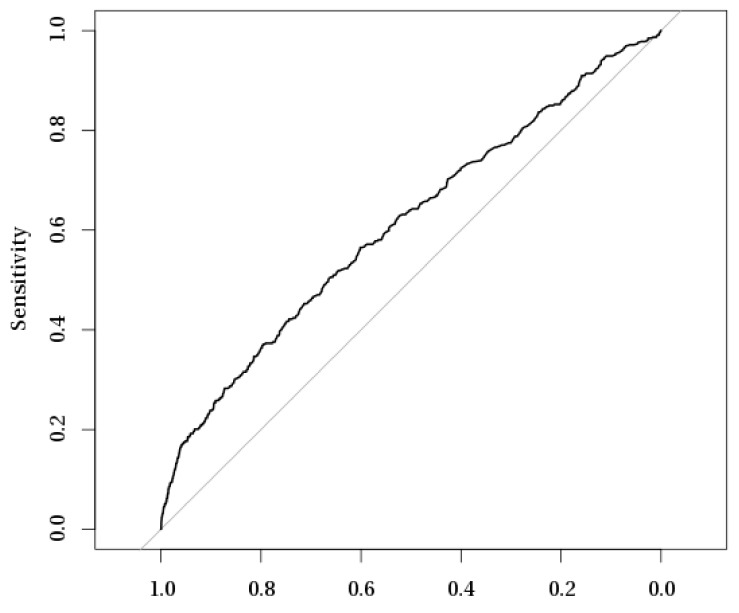
Receiver operating characteristic (ROC) curve analysis of sensitivity/specificity evaluation of multiple logistic regression model for chronic postoperative endophthalmitis.

**Table 1 ijerph-16-02188-t001:** The total number of postoperative endophthalmitis (POE) cases and the total number of cataract surgeries performed in Poland from 2010–2015 matched with population data by age group.

	2010	2011	2012	2013	2014	2015
No. of 0–18 year-olds	7,643,553	7,630,880	7,531,582	7,431,731	7,367,066	7,309,001
No. of cataract surgeries	245	267	272	294	270	242
No. with endophthalmitis	0	0	0	0	0	0
No. of 19–39 year-olds	12,482,309	12,523,386	12,461,398	12,355,235	12,201,430	12,015,345
No. of cataract surgeries	1441	1305	1280	1324	1383	1384
No. with endophthalmitis	5	5	3	10	8	4
No. of 40–49 year-olds	4,792,211	4,822,159	4,838,436	4,879,816	4,956,005	5,064,587
No. of cataract surgeries	3007	2680	2620	2583	2849	2920
No. with endophthalmitis	7	9	4	6	6	4
No. of 50–59 year-olds	5,770,823	5,765,460	5,656,651	5,536,118	5,406,320	5,245,352
No. of cataract surgeries	15,598	13,397	13,203	13,188	14,645	14,001
No. with endophthalmitis	11	27	20	22	19	18
No. of 60–69 year-olds	3,682,048	3,931,289	4,171,206	4,409,809	4,642,821	4,888,294
No. of cataract surgeries	38,973	35,322	37,986	43,120	52,747	57,646
No. with endophthalmitis	46	43	49	59	53	42
No. with age ≥70 years	4,146,056	3,852,826	3,874,727	3,889,291	3,910,358	3,932,421
No. of cataract surgeries	141,819	122,035	126,644	134,212	156,970	160,905
No. with endophthalmitis	183	160	145	154	120	89
Total No.	38,517,000	38,526,000	38,534,000	38,502,000	38,484,000	38,455,000
No of cataract surgeries	201,083	175,006	182,005	194,721	228,864	237,098
No. with endophthalmitis	252	244	221	251	206	157

**Table 2 ijerph-16-02188-t002:** Incidence of acute and chronic postoperative endophthalmitis (POE) after cataract surgery in Poland from 2010–2015.

Year	No. of Cataract Surgeries	No. of Cases of Acute POE	Incidence of Acute POE (%)	No. of Cases of Chronic POE	Incidence of Chronic POE (%)	Total Number of Cases of POE	Total Incidence of POE (%)
2010	201,083	95	0.047	157	0.078	252	0.125
2011	175,006	95	0.054	149	0.085	244	0.139
2012	182,005	97	0.053	124	0.068	221	0.121
2013	194,721	118	0.061	133	0.068	251	0.129
2014	228,864	96	0.042	110	0.048	206	0.09
2015	237,098	83	0.035	74	0.031	157	0.066
Total	1,218,777	584	0.048	747	0.061	1331	0.109

*R*-tests for data proportions: *p* = 0.0018; *p* = 0.0147; *p* = 0.0000.

**Table 3 ijerph-16-02188-t003:** Demographic characteristics of all postoperative endophthalmitis (POE) cases after cataract surgery in Poland during 2010–2015.

	Acute POE *n* (%); Rate Per 1000 Surgeries	Chronic POE *n* (%); Rate Per 1000 Surgeries	Total Number of POE *n* (%); Rate Per 1000 Surgeries
Age:			
0–18	0 (0.00%); 0.000	0 (0.00%); 0.000	0 (0.00%); 0.000
19–39	22 (3.77%); 2.710	13 (1.74%); 1.601	35 (2.63%); 4.311
40–49	16 (2.74%); 0.960	20 (2.68%); 1.200	36 (2.70%); 2.160
50–59	51 (8.73%); 0.607	66 (8.83%); 0.785	117 (8.79%); 1.392
60–69	125 (21.40%); 0.470	167 (22.36%); 0.628	292 (21.94%); 1.098
70+	370 (63.36%); 0.439	481 (64.39%); 0.570	851 (63.94%); 1.009
Sex			
Women	328 (56.16%); 0.413	440 (58.90%); 0.554	768 (57.70%); 0.967
Men	256 (43.84%); 0.602	307 (41.10%); 0.773	563 (42.30%); 1.325
Urban residence	408 (69.86%); 0.454	539 (72.15%); 0.600	947 (71.15%); 1.054
Rural residence	176 (30.14%); 0.568	208 (27.85%); 0.671	384 (28.85%); 1.239
Diabetes mellitus t. I	24 (4.11%); 0.248	35 (4.69%); 0.361	59 (4.43%); 0.609
Diabetes mellitus t. II	99 (16.95%); 0.399	151 (20.21%); 0.609	250 (18.78%); 1.008
No diabetes mellitus	461 (78.94%); 0.527	561 (75.10%); 0.642	1022 (76.79%); 1.169

**Table 4 ijerph-16-02188-t004:** Surgical characteristics of all postoperative endophthalmitis (POE) cases after cataract surgery in Poland during 2010–2015.

	Acute POE *n* (%)	Chronic POE *n* (%)	Total Number of POE *n* (%)
Cataract surgery in multidisciplinary hospital	559 (95.72%)	704 (94.24%)	1263 (94.89%)
Cataract surgery in non-multidisciplinary hospital	25 (4.28%)	43 (6.76%)	68 (5.11%)
Surgical technique phacoemulsification	561 (96.06%)	691 (92.50%)	1252 (94.06%)
Surgical technique extracapsular extraction	23 (3.94%)	56 (7.50%)	79 (5.94%)
One-day cataract surgery	219 (37.50%)	215 (28.78%)	434 (32.61%)
Non one-day cataract surgery	365 (62.50%)	532 (71.22%)	897 (67.39%)
Cataract surgery combined with corneal transplantation	0 (0.00%)	16 (2.14%)	16 (1.20%)
Cataract surgery combined with glaucoma filtration surgery	4 (0.68%)	16 (2.14%)	20 (1.50%)
Cataract surgery combined with pars plana vitrectomy	71 (12.16%)	66 (8.84%)	137 (10.30%)
Non-combined cataract surgery	509 (87.16%)	649 (86.88%)	1158 (87.00%)

**Table 5 ijerph-16-02188-t005:** Multiple logistic regression models of the risk factors for acute and chronic postoperative endophthalmitis (POE).

Variables	Acute POE OR, 95% CI, *p*-value	Chronic POE OR, 95% CI, *p*-value
Age, per year increase	0.99 (0.98–1.00); *p* = 0.047	1.00 (0.99–1.01); *p* = 0.454
Men vs. women	1.35 (1.11–1.64); *p* = 0.003	1.28 (1.08–1.53); *p* = 0.005
Rural residence	1.22 (0.98–1.50); *p* = 0.069	0.98 (0.80–1.18); *p* = 0.811
Type II diabetes mellitus	1.18 (0.91–1.52); *p* = 0.198	1.42 (1.18–1.75); *p* = 0.001
Extracapsular extraction	1.40 (0.77–2.31); *p* = 0.226	3.09 (2.20–4.23); *p* = 0.000
One-day surgery	1.11 (0.90–1.36); *p* = 0.323	0.75 (0.61–0.91); *p* = 0.004
Combined cataract surgery	4.25 (3.09–5.74); *p* = 0.000	4.41 (3.38–5.67); *p* = 0.000
Surgery in a non-multidisciplinary hospital	0.77 (0.47–1.17); *p* = 0.246	0.87 (0.58–1.24); *p* = 0.474

OR = odds ratio; CI = confidence interval.

**Table 6 ijerph-16-02188-t006:** Comparison of the incidence rates of postoperative endophthalmitis (POE) from recent studies which comprised over 100,000 participants.

Epidemiological Study	Total Number of Cataract Extractions	Time Period (Years)	Total Incidence of POE *n* (%)	Incidence of Acute POE ^†^*n* (%)	Incidence of Chronic POE ^‡^*n* (%)	Incidence Reduction From (%)–(%)
Nationwide Study in France (France) [7]	6,371,242	2005–2014	6668 (0.105%)	6668 (0.105%)	Not applicable (N/A)	0.145%–0.053%
Medicare Database of Cataract Surgery (USA) [13]	3,280,966	2003–2004	4,006 (0.122%)	4,006 (0.122%)	N/A	0.132%–0.111%
National Database of Hospitalizations—current study from Poland	1,218,777	2010–2015	1331 (0.109%)	584 (0.048%)	747 (0.061%)	0.125%–0.066%
Quebec State Control for Health Insurance (Canada) [8]	490,690	1996–2005	754 (0.154%)	754 ^§^ (0.154%)	N/A	2.1%–0.8% ^β^
Farabi Eye Hospital (Iran) [2]	480,104	2006–2014	112 (0.023%)	100 (0.021%)	12 (0.002%)	N/A
Swedish National Study (Sweden) [6]	464,996	2005–2010	135 (0.029%)	135 (0.029%)	N/A	0.03%–0.02%
Ontario Health Insurance Plan (Canada) [9]	442,177	2002–2006	617 (0.139%)	617 ^Γ^ (0.139%)	N/A	N/A
Kaiser Permanente (USA) [16]	315,246	2005–2012	215 (0.068%)	215 ^§^ (0.068%)	N/A	N/A
The Malaysian Cataract Surgery Registry (Malaysia) [14]	163,503	2008–2014	131 (0.08%)	131 (0.08%)	N/A	0.11%–0.08%
Aravind Eye Hospital (India) [17]	116,714	2014–2015	65 (0.056%)	65 (0.065%)	N/A	N/A

^†^ Acute endophthalmitis was identified if the symptoms occurred within 1–42 days from cataract surgery; ^‡^ chronic endophthalmitis if the symptoms occurred ≥43 days after cataract surgery; ^§^ acute endophthalmitis was identified if the symptoms occurred within 1–90 days from cataract surgery; ^Γ^ acute endophthalmitis was identified if the symptoms occurred within 1–14 days from cataract surgery; ^β^ rate per 1000 cataract surgical procedures.

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
