# Peer review of "Incidence and Characteristics of Endophthalmitis after Cataract Surgery in Poland, during 2010–2015"

_ijerph, 2019, doi:10.3390/ijerph16122188_

Round 1

Reviewer 1 Report

The manuscript is well written, I have the following points for the authors to address:

1.      Table 1: Column title low should be put at the top low of the table.

       Table 3: Row titles, Diabetes Mellitus E10 and E1, should be changed to Diabetes Mellitus                             type 1 and type 2, respectively. The word “No” for No Diabetes Mellitus is                                      misallocated at wrong low.

        Table 4: Column title low should be put at the top low of the table. There are misalignments                        of row titles.

2.      The manuscript contains no conclusion section.

3.      I would suggest the authors add an analysis to find the possible explanation why the acute            and chronic POE decrease over years.  For example, exam any associated changes of the             surgical characteristics or risk factors with POEs over years.

Author Response

Dear Reviewer,

 We would like to thank you and other reviewers for your kind, friendly and instructive comments on our paper. Following your suggestions we have made some corrections. Please find my responses and list of the changes along with modified manuscript as an attached file. 

Yours sincerely

            Michal S. Nowak MD, PhD 

Reviewer 1

Comments to the Author:

The manuscript is well written, I have the following points for the authors to address:

1.      Table 1: Column title low should be put at the top low of the table.

Table 3: Row titles, Diabetes Mellitus E10 and E1, should be changed to Diabetes Mellitus type 1 and type 2, respectively. The word “No” for No Diabetes Mellitus is misallocated at wrong low.

Table 4: Column title low should be put at the top low of the table. There are misalignments of row titles.

Ad.1 Yes, we agree with this comment. We have changed Table 1, Table 3 and Table 4 according to your suggestions. We have also modified Table 1 according to the suggestions of Reviewer 3 and Table 3 according to the suggestions of Reviewer 2.  Please see, in the modified manuscript.

2.      The manuscript contains no conclusion section.

Ad.2 Yes, we agree with this comment. In the current shape, this could make some confusions. According to your suggestion, we have added the conclusions section. Please see, in the modified manuscript: page 11, lines 242-249

3.      I would suggest the authors add an analysis to find the possible explanation why the acute and chronic POE decrease over years.  For example, exam any associated changes of the surgical characteristics or risk factors with POEs over years.

Ad.3 Yes, we agree with this comment. According to your suggestion we have analyzed the surgical characteristics of cataract surgery in the study period and we have added following sentence to the discussion section:

“We analyzed the surgical characteristic of cataract surgery in Poland to find the possible explanation why the acute and chronic POE decrease over years. We found that during the study period the total number of one-day cataract surgeries significantly increased while the use of extracapsular lens extraction significantly decreased.[10]”

Please see, in the modified manuscript: page 9, line 205 and page 10 lines 206-207

We have made some other changes according to others Reviewers suggestions and uploaded the modified manuscript.

Reviewer 2 Report

Article “Incidence and characteristics of endophthalmitis after cataract surgery in Poland, during 2010-2015” deals with very important complication after cataract surgery - acute and chronic postoperative ophthalmitis as it is the most serious and vision threatening post surgical complication. Data about incidence of postoperative endophthalmitis is lacking especially in East European countries. 

Comments:

Table 1 - first line with study years is presented in the middle of the table, the table starts  with elder age groups.

Table 2 deals with incidence of acute and chronic postoperative endophthalmitis. The incidence  decreased from 2010 to 2015, but gradually increased in the period 2010-2013.  Year 2014 and 2015 - decrease in acute and total, 2015 - also in chronic. The is not discussed in the results and the statistical significance of change is not mentioned.

Table 3. Presenting rate of endophthalmitis in each age group could be more informative than presenting distribution of endophthalmitis cases according to age. Cataract surgery volume is increasing with age, so the number of endophthalmitis cases  as well. also -  presenting rate of endophthalmitis according to sex, urban/rural area and diabetes.

Limitations are described as possible misclassification and lacking  data about use of intracameral antibiotics but not the collecting data about all possible risk factors for postoperative endophthalmitis.

Intraoperative complications as important risk factor for postoperative endophthalmitis was found in other studies (this is mentioned in discussion, line 198), but were not evaluated in this study, type of IOL used - it could be also the limitation of this study.

Section 5 - conclusions - is not found.

Author Response

Dear Reviewer,

 We would like to thank you and other reviewers for your kind, friendly and instructive comments on our paper. Following your suggestions we have made some corrections. Please find my responses and list of the changes along with modified manuscript as an attached file. 

Yours sincerely

            Michal S. Nowak MD, PhD 

Reviewer 2
Comments to the Author:

 Article “Incidence and characteristics of endophthalmitis after cataract surgery in Poland, during 2010-2015” deals with very important complication after cataract surgery - acute and chronic postoperative ophthalmitis as it is the most serious and vision threatening post surgical complication. Data about incidence of postoperative endophthalmitis is lacking especially in East European countries. Comments:

Table 1 - first line with study years is presented in the middle of the table, the table starts  with elder age groups.

Ad.1 Yes, we agree with this comment. We have changed Table 1 according to yours suggestions. We have also modified Table 1 according to the suggestions of Reviewer 3. Please see, in the modified manuscript.

2.      Table 2 deals with incidence of acute and chronic postoperative endophthalmitis. The incidence  decreased from 2010 to 2015, but gradually increased in the period 2010-2013.  Year 2014 and 2015 - decrease in acute and total, 2015 - also in chronic. The is not discussed in the results and the statistical significance of change is not mentioned.

Ad. 2 Yes, we agree with this comment. We have added the statistical analyses to the Table 2. We have also discussed it in the results and in the discussion sections. The following sentences have been added: please see, in the modified manuscript: page 4, lines 116-119

“The differences between the incidences of acute, chronic and the total number of POE in years 2010-2015 were statistically significant (p=0,0018, p=0,0147 and p=0,0000 respectively) with a temporal increase in the incidence of POE in years 2011-2013.”

and page 11, lines 230-231

 “This cost reduction was also a possible factor of temporal increase in the incidence of POE in years 2011-2013. “

3.      Table 3. Presenting rate of endophthalmitis in each age group could be more informative than presenting distribution of endophthalmitis cases according to age. Cataract surgery volume is increasing with age, so the number of endophthalmitis cases  as well. also -  presenting rate of endophthalmitis according to sex, urban/rural area and diabetes.

Ad. 3 Yes, we agree with this comment. We have changed Table 3 according to your suggestions. Please see, in the modified manuscript.

We have also added following sentence to the results section. Please see: page 4, line 131 and page 5, line 132

„Young males had relatively more POE probably due to traumatic cataract.”

4.      Limitations are described as possible misclassification and lacking  data about use of intracameral antibiotics but not the collecting data about all possible risk factors for postoperative endophthalmitis. Intraoperative complications as important risk factor for postoperative endophthalmitis was found in other studies (this is mentioned in discussion, line 198), but were not evaluated in this study, type of IOL used - it could be also the limitation of this study.

Ad. 4. Yes, we agree with this comment. It would be better to include data on the types of IOL used in the cataract surgery in Poland. However, these data are currently not collected in the national database of hospitalizations. We have modified the discussion section according to yours suggestions. Please see, in the modified manuscript: page 11, lines 237-241

“During the study period, the use of intracameral antibiotic injection at the end of cataract surgery was not officially reported in the NFZ national database of hospitalizations as well as other possible risk factors like surgical complications and the type of IOL used. However, the cost-effectiveness of the use of intracameral antibiotics in reduction of the substantial costs associated with the treatment of POE after cataract surgery is well known. [22]”

5.      Section 5 - conclusions - is not found.

Ad.5 Yes, we agree with this comment. In the current shape, this could make some confusions. According to your suggestion, we have added the conclusions section. Please see, in the modified manuscript: page 11, lines 242-249.

We have made some other changes according to others Reviewers suggestions and uploaded the modified manuscript.

Reviewer 3 Report

The paper is not finished. Table 1 has numbers that do not correspond to anything on the paper. Conclusions and Author Contributions are missing.

Please upload the final version of the article.

Author Response

Dear Reviewer,

 We would like to thank you and other reviewers for your kind, friendly and instructive comments on our paper. Following your suggestions we have made some corrections. Please find my responses and list of the changes along with modified manuscript as an attached file. 

Yours sincerely

            Michal S. Nowak MD, PhD 

Reviewer 3
Comments to the Author:

The paper is not finished. Table 1 has numbers that do not correspond to anything on the paper. Conclusions and Author Contributions are missing.

Ad.1 Yes, we agree with this comment. In the current shape, this could make some confusions. According to your suggestions, we have modified Table 1.

We have also added the conclusions section and Author Contributions.

We have made some other changes according to others Reviewers suggestions and uploaded the modified manuscript.

Round 2

Reviewer 3 Report

The major issues of the article have been solved. I am content with the article.